# High-Order Harmonic Film Bulk Acoustic Resonator Based on a Polymer Reflector

**DOI:** 10.3390/s22197439

**Published:** 2022-09-30

**Authors:** Yuxuan Hu, Bo Dong, Liang Lei, Zhizhong Wang, Shuangchen Ruan

**Affiliations:** College of Integrated Circuits and Optoelectronic Chips, Shenzhen Technology University, Shenzhen 518118, China

**Keywords:** FBAR, polymer, high-order resonance, FEM

## Abstract

A film bulk acoustic resonator (FBAR), based on a polymer air cavity, is presented. The polymer reflective layer on the polymer air cavity can serve both as the reflective layer and the function layer for inducing the high-order mode resonance. With the aluminum nitride as the piezoelectric layer, the resonance frequency of the FBAR can reach 6.360 GHz, based on the finite element method. The product of the corresponding frequency and the quality factor, f × Q is more than 2 × 10^13^. This design model provides a good solution for the high-frequency filters and high-sensitivity sensor designs.

## 1. Introduction

The popularization of fifth-generation communication equipment and the development of the internet of things has made the internet of everything possible, and these are inseparable from bulk acoustic wave (BAW)/surface acoustic wave (SAW) resonators. Due to the structural limitations of the SAW resonator, the higher the operating frequency, the thinner the interdigitated electrodes [1], and the thinner electrodes are difficult to withstand high frequency and power. The BAW filters are more frequently used in high-frequency communication and sensing because of their higher power tolerance. The BAW filters are divided into film bulk acoustic resonator (FBARs) [2,3,4], solidly mounted resonators (SMRs) [5,6], and laterally-excited bulk-wave resonators (XBARs) [7,8,9], based on the structure difference. At present, the piezoelectric film materials used in the BAW resonators mainly include aluminum nitride, zinc oxide, and PZT. Compared with the polycrystalline piezoelectric crystals, the single crystal piezoelectric crystals do not cause the scattering of acoustic waves inside the piezoelectric crystals, which helps to increase the speed of the acoustic wave. Growing high-performance piezoelectric films is key to the BAW fabrication, the common methods are physical vapor deposition (PVD) and chemical vapor deposition (CVD) [10,11]. For the resonator, due to the design deviation, the BAWs propagating in the film are accompanied by many parasitic modes, while coupling-out spurious modes often affect the signal-to-noise ratio of the filtered signal.

Currently. the known resonant modes of the FBAR include three modes: TE, TS, and E: (1) TE-FBAR is the earliest developed and the most widely used type on the market. Its c-axis is along the normal direction of the film plane (the thickness direction). It is mainly used in filters, duplexers, and oscillators. It is used in radio frequency devices such as communication systems [12,13]. (2) The FBAR (TS-FBAR) with the basic thickness-shear (TS) as the resonance mode, needs a lining process to make the piezoelectricity during the material processing. The axis forms an inclination angle with the excitation electric field in the limit direction of the piezoelectric thin acid circuit, which excites the vibration response dominated by the TS mode. The TS-FBAR can maintain a high-quality factor when in contact with liquid and is usually used as a sensor in liquid or for biochemical information detection [14,15,16,17]. (3) The FBAR with in-plane stretching (E) as the resonance mode is usually called a contour-mode FBAR, which is also realized by tilting the c-axis of the piezoelectric film. Its main feature is that the resonant frequency of the device is greatly affected by the lateral size of the electrode, so it has inherent advantages in the integrated application of the device [18,19,20]. In recent years, more and more researchers have begun to study higher frequency resonators, based on the parasitic modes and higher-order resonances [21,22,23]. Regardless of the modes, they prevent acoustic waves from leaking into the substrate.

Here, a new FBAR design, based on a polymer air cavity is presented. The polymer reflective layer of the cavity serves not only as the high reflectivity layer of the FBAR but also as the function layer for exciting the high order mode resonance. The results show that the polymer reflective layer in the FBAR has a stronger resonance on both sides of the fundamental mode than that of the traditional FBAR. With the aluminum nitride as the piezoelectric layer, the resonance frequency of the FBAR can reach 6.360 GHz. The product of the corresponding frequency and the quality factor, f × Q is more than 2 × 10^13^. This design simplifies the fabrication process of the FBAR since a simple photolithography process can create this polymer cavity and avoid the complicated etching process of the traditional air cavity. Moreover, this design introduces the new characteristics of the FBAR. It is expected to have practical applications in the FBAR based filters and sensors.

## 2. Principle of the Polymer-FBAR

In recent years, researchers have proposed many different simulation analysis methods, such as the Mason and Butterworth Van Dyke one-dimensional simulation, and the simulation results are consistent with the experimental results [24,25].Therefore, it has become one of the most popular simulation methods. Nevertheless, the one-dimensional modeling simulation is not enough to pay attention to the stray mode in the simulation of the FBAR resonator. The acoustic wave propagates inside the FBAR, and the electromechanical coupling in the transverse and longitudinal directions exists at the same time, so it is necessary to carry out a two-dimensional finite element simulation of the FBAR resonator. Multiphysics systems can be represented by the boundary conditions and a set of equations. The stress equation of the motion can be expressed as [26]:(1)$$T_{ij}=\rho u_{j},$$
where *T_ij_* is the stress matrix, ***ρ*** is the density of the piezoelectric materials, and *u_j_* is displacement. The electrostatic Gaussian equation can be expressed as [27]:(2)$$D_{i,j}=0.$$

The following stress-charge equations describes the electrostrictive behavior of the piezoelectric materials and it can be solved using simulation software. These equations are one of many forms that describe the inverse piezoelectric effect and the direct piezoelectric effect [28,29,30,31],
(3)$$T_{ij}=c_{ijkl}\cdot S_{kl}-e_{kij}\cdot E_{k}$$
(4)$$D_{i}=e_{ikl}\cdot S_{kl}+\varepsilon_{ik}\cdot E_{k}$$
where *c_ijkl_* is the elastic constant matrix, *e_kij_* is the piezoelectric constant matrix, *e_kij_* is the transpose of *e_ikl_*, *ε* is the dielectric constant matrix, *S* is the strain, *E* is the electric field, and *i*, *j*, *k*, *l* are tensor indices.

The direct analysis of the results for the multiphysics systems is quite difficult. The finite element method can solve the multidimensional complex systems by using sophisticated approximation techniques for partial differential equations and discretizing the geometry of the system [32,33]. COMSOL^®^ Multiphysics 5.6 (Comsol Multiphysics GmbH, Göttingen, Germany) is an important tool for solving such complex systems.

The piezoelectric conversion efficiency is usually described by the electromechanical coupling coefficient, but the difference between the electromechanical coupling coefficient *K*^2^ and the effective electromechanical coupling coefficient *K_teff_^2^* should be noted. *K*^2^ is the electromechanical coupling coefficient of the piezoelectric material, which is the intrinsic characteristic of the material, and *K_teff_*^2^ is the effective electromechanical coupling coefficient of the resonator, which is not only related to the piezoelectric material, but also related to various resonator designs, such as film thickness, film material, and resonator structure. The electromechanical coupling coefficient *K*_2_ of the material can be obtained from the blew formula:(5)$${K_{teff}}^{2}=\frac{\left(\frac{\pi }{2}\right)\left(\frac{f_{r}}{f_{a}}\right)}{\tan\left(\left(\frac{\pi }{2}\right)\left(\frac{f_{r}}{f_{a}}\right)\right)}\approx {\left(\frac{\pi }{2}\right)}^{2}\left(\frac{f_{a}-f_{\gamma }}{f_{a}}\right)$$
where *f_r_* is the resonant frequency and *f_a_* is the anti-resonant frequency [34,35,36]. The FBAR devices can use the piezoelectric effect to generate the mechanical resonance under the action of an input electrical signal, and correspondingly, the mechanical resonance can also be converted into an electrical signal output.

Figure 1 shows our proposed polymer FBAR 3D model. It has six parts, including the substrate, the polymer air cavity, the support layer, the bottom electrode layer, the piezoelectric layer, and the top electrode layer.

The reflectance formula for the FBAR structure can be given by:(6)$$c=\frac{Z_{1}-Z_{2}}{Z_{1}+Z_{2}}$$
where *C* is the reflection coefficient, *Z*_1_ is the characteristic acoustic impedance of the support layer, and *Z*_2_ is the characteristic acoustic impedance of the polymer [37]. The characteristic acoustic impedance of a material is equal to the product of the density of the material and the velocity of the acoustic wave of the material. The characteristic acoustic impedance of the bottom electrode and the support layer is large while that of the characteristic acoustic impedance of the reflective polymer layer is small, resulting in a large reflection at the interface of the polymer support layer (*C* > 90%) [38]. Thin polymer layers cannot withstand strong stress, adding a silicon nitride support layer enhances the mechanical stability of the polymer FBAR [39,40,41].

A simulation analysis is performed by the finite element method (FEM) supported by COMSOL^®^ Multiphysics 5.6 software. The proposed model of the polymer FBAR in the finite element simulation environment is shown in Figure 2, and the 3 μm perfectly matched layers (PML) on both sides and the bottom of the substrate are adopted. The bottom of the substrate is set as a fixed constraint, the drive voltage or power is applied to the top electrode and to the ground on the bottom electrode, and the frequency domain is used to calculate the impedance spectrum and reflection coefficient of the polymer FBAR.

In the finite element simulation environment, the traditional FBAR model and the polymer-FBAR model are established, respectively. The traditional FBAR has a silicon based air cavity while the proposed FBAR has a polymer air cavity. All parameters are the same except for the polymer layer for comparing the performance difference between the traditional FBAR and polymer-FBAR, from their results.

As shown in Figure 3, the traditional FBAR has only one resonance peak while two strong resonance peaks on both sides of the resonance peak of the traditional cavity FBAR are induced, for the polymer FBAR. Moreover, the resonance impedance ratio of the polymer FBAR can reach 52 dB. The resonator based on this resonance mode can work at a higher frequency resonance than that of the traditional FBAR. Unlike the crystalline materials, the spatial lattice of the polymer materials is not periodic with the strong scattering of acoustic waves, so a variety of eigenmodes are coupled with each other to form complex coupled vibration modes. In addition, the acoustic energy is suppressed inside the aluminum nitride piezoelectric layer to achieve a more efficient electromechanical coupling, reducing the leakage of the acoustic waves into the substrate material which may lead to insertion loss.

## 3. Results and Discussions

### 3.1. Electrode

The simulation model AlN is selected as the piezoelectric layer with the thickness of 1 μm, and the thickness of the electrode is set to 0.1 μm. The polymer layer is a polyimide material with a thickness of 1 μm. Due to the low mechanical strength of the polymer layer, it can only provide the effect of bulk acoustic wave coupling. Hence, it is necessary to support the electrode-piezo-electrode sandwich structure. The Si_3_N_4_ materials with a higher mechanical strength is used as the support layer with the thickness of 1 μm. The top electrode is rectangular and the irregular pentagonal electrode suppresses the coupling of the parasitic modes and weakens the high-order resonances [42]. A model was established in COMSOL^®^ Multiphysics 5.6 to simulate the effect of the electrodes of different materials on the intensity of the high-order resonance peak.

Figure 4a, shows the absolute value of the admittance for the electrode materials at different resonance frequencies. With the increase of the acoustic impedance, the magnitude of the first and the second resonance admittance decreases accordingly. Figure 4b shows the relationship between the admittance amplitude and the acoustic impedance of the electrodes. The metal electrodes with a higher density of Au, Ag, and Pt have a lower resonance strength, since they have a higher acoustic impedance which leads to a higher bulk acoustic wave reflection at the interface between the electrode and piezoelectric film. The electrode material with the lowest density is Al. Hence, it has the highest admittance at the resonance frequencies. Figure 4c shows that the second-order mode reflection coefficient of Al is the largest, and the Q factor is calculated by S11. The Q factor is Al 2121, Ti 1398, Mo 2021.

### 3.2. Polymer Materials

There are many kinds of polymer materials, and Poisson’s ratio of Young’s modulus varies greatly among the different polymer materials. The acoustic vibration coupled by the piezoelectric film is transmitted to the polymer layer, and within the elastic limit, the high-order vibration mode of the piezoelectric layer can be strengthened or weakened. Four different polymer materials are selected for the simulation, namely: polyimide (PI), polyethylene (PE), polymethyl methacrylate (PMMA) and polyamide (PA). The mechanical parameters are provided by COMSOL^®^ Multiphysics 5.6, as shown in Table 1.

Figure 5a shows the absolute value of the polymer materials’ admittance response at the different frequencies. As can be seen, the FBAR with the polyimide reflective layer has a higher resonance frequency. Since the PI has high thermal stability, a high insulation, and a high mechanical strength, it is an ideal polymer material for the polymer reflective layer FBAR. Figure 5b shows the S11 reflection spectra of the second-order resonance mode of the polymer FBAR. It can be seen that the PI FBAR has the highest resonance frequency. Its resonance frequency can reach 6.360 GHz. A variety of resonances coupled out because the polymer does not have a strict crystal structure, which helps the polymer FBAR to induce the higher frequency resonance.

### 3.3. Polyimide Thickness

Figure 6 shows the absolute admittance of the polymer FBAR under different polyimide thicknesses. It can be seen that the resonance intensity and frequency of the second resonance peak of the PI FBAR under a thickness of 1μm, are higher than those of the other polymer thicknesses. The polyimide with a thickness between 1.0–1.4 μm, can couple out the second high-order resonance. When the thickness of the polymer layer increases to 2 μm, the second-order high-order strong resonance disappears. Moreover, for the polyimide films with a thickness of less than 1 μm, there is no second high-order resonance in the admittance spectrum. In an ideal fluid, there is no shear deformation and the coefficient of the viscosity is zero, so the medium has only a volumetric deformation in which only the compressional waves propagate. However, in the solid medium, in addition to the volume deformation, the shear deformation will be generated. Therefore, the deformation of the solid medium will generate two kinds of waves, namely, the compression wave and the shear wave. Therefore, when the bulk acoustic wave is incident into the polyimide film, multiple modes of acoustic waves are coupled, resulting in modes higher than the fundamental frequency.

### 3.4. Support Layer Thickness

The absolute admittance spectra under the different thicknesses of the Si_3_N_4_ support layer are shown in Figure 7. The admittance response amplitude and the frequency are higher as the thickness of the Si_3_N_4_ layer is between 0.7 μm and 1.4 μm, and the speed of the bulk acoustic wave is Si_3_N_4_ is 9000 m/s, and the wavelength of the acoustic wave under the frequency of 6.360 GHz propagating in the support layer is 1.4 μm, which is thinner than the half wavelength and maty induce, the acoustic wave coupling of the surface waves and shear waves. It should be noted that if the supported layer is thicker than the full wavelength, the longitudinal waves coupled out of surface waves and shear waves may be induced to cause a large loss of acoustic energy.

## 4. Discussion

In micro electric mechanical systems (MEMS), the selection of electrodes often has a great impact on the performance of the device. For example, Au and Pt have a high stability, therefore, it is mostly used in the more extreme external environments [43]. The melting point of Ir is 2450 °C, so Ir electrodes are often used in high-temperature devices [44]. However, for the polymer-FBAR, it is not only necessary to return the acoustic energy to the piezoelectric layer in order to form resonance, but it also requires the mechanical vibration coupling of the acoustic waves in the polymer layer, so the Al and Ti electrode materials with a low acoustic impedance are conducive to the leakage of acoustic waves into the polymer layer, forming complex multi-order coupling modes, and enhancing the high-order polymer FBAR resonance. 

The FBAR is a high-frequency device. Under a high-frequency operation, heat will be generated due to the inherent loss of the material. Therefore, the high-temperature resistance of the polymer material will seriously affect the temperature stability of the FBAR. The PI is a better polymer material for high-temperature chips because of its high-temperature stability. The PI has good dielectric properties, and its dielectric constant is about 3.4, and can be reduced to about 2.5 by introducing fluorine or dispersing air nanometer size in polyimide. Its dielectric loss is 10^−3^, its dielectric strength is 100–300 kV/mm, and its volume resistance is 10^17^ Ω·cm. It should be noted that the polymer film used in this simulation is polyimide film and it has stable thermal characteristics. Hence, the proposed FBAR has a stable frequency response to temperature. A good crystal structure can be grown under extreme growth conditions without damaging the polyimide substrate [45].

Based on Stokes’ law, the amplitude attenuation of the elastic waves in a homogeneous medium and the x-direction [46]:(7)$$A\left(x\right)=A_{0}\sin\left(\omega t\right)\exp\left(-ax\right),$$
where *A_0_* is the incident sine wave amplitude, *α* is the attenuation coefficient, which is greatly affected by the ***ω*** and temperature, as the temperature increases and the incident sine wave *ω* increases, the prompt attenuation coefficient sharply increases. The thin polyimide film is easy to leak the acoustic wave into the substrate, which induces the larger loss of the bulk acoustic wave, and the thicker polyimide film can suppress the generation of high-order resonance. Hence, the optimized polymer film thickness is needed.

The Si_3_N_4_ film has a relatively regular crystal structure so the loss of the acoustic wave energy is small. The analysis of the effect of Si_3_N_4_ thickness on the mode coupling in the FBAR of this study follows the resonator mode coupling and the characteristic dispersion of the SMR structure [10].The Si_3_N_4_ film with a thickness of less than 0.5 μm reduces the mechanical q value of the FBAR, which is not conducive to the mechanical stability of the FBAR; The high-order resonance reinforced by the film layer needs to be reflected back to the piezoelectric layer, to form resonance. In addition, the Si_3_N_4_ film larger than 1.5 μm will scatter this part of the sound wave, thus weakening the secondary resonance strength.

## 5. Conclusions

In summary, a novel polymer FBAR design is proposed. The effect of the polymer material, the electrode material, the thickness of the polymer layer, and the thickness of the support layer are optimized by using the finite element simulation method. It is found that the acoustic impedance of the heavy electrode material weakens the second resonance strength; the resonator with 1 μm PI polymer layer has a higher second resonance frequency; the Si_3_N_4_ support layer with a 1 μm thickness has a higher second resonance strength and a higher frequency. The research in this paper guides the design of the polymer FBAR with the polymer air cavity. In addition, the polymer air cavity acoustic wave resonators have a great potential in the direction of flexible wearable FBAR sensors.

## Figures and Tables

**Figure 1 sensors-22-07439-f001:**
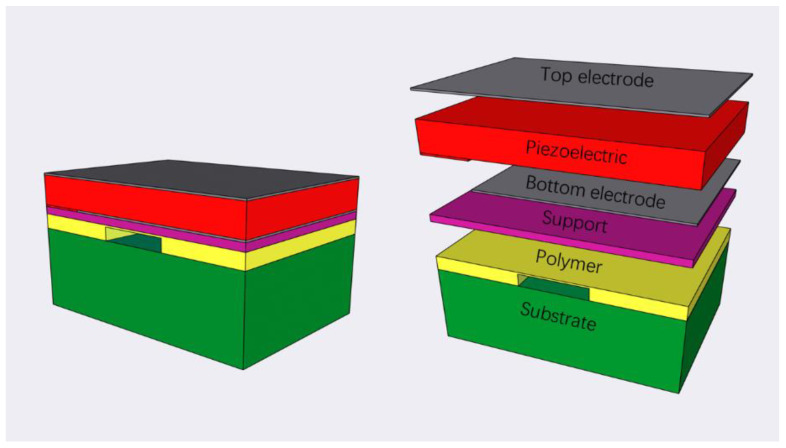
3D model of the FBAR resonator.

**Figure 2 sensors-22-07439-f002:**
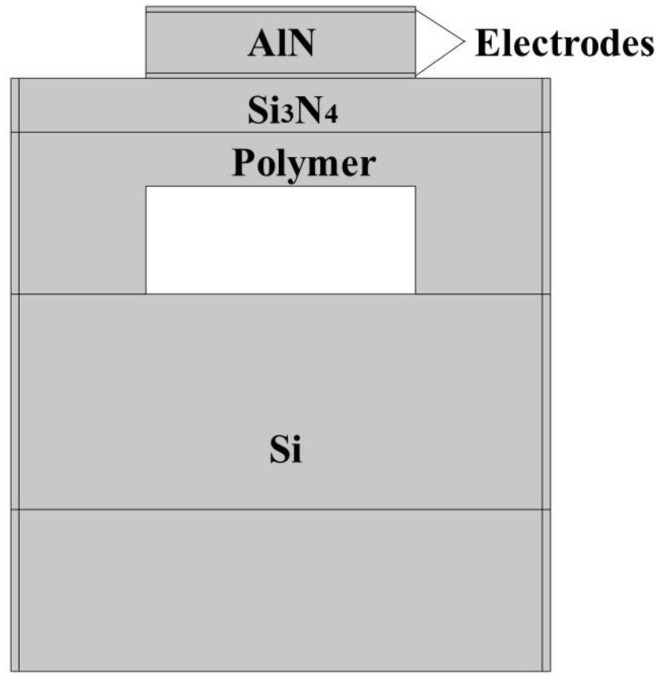
Polymer-FBAR finite element simulation model.

**Figure 3 sensors-22-07439-f003:**
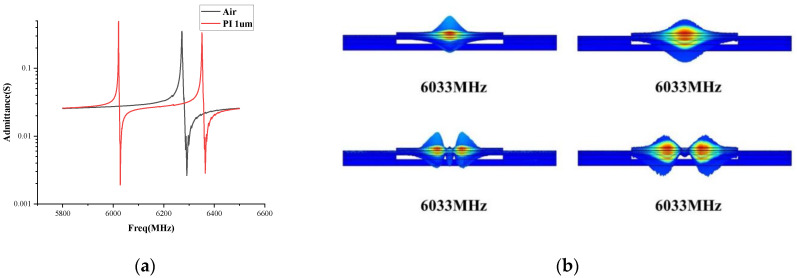
(**a**) Absolute admittance curves for the conventional FBAR and polymer FBAR; (**b**) Resonance and antiresonance energy distribution.

**Figure 4 sensors-22-07439-f004:**
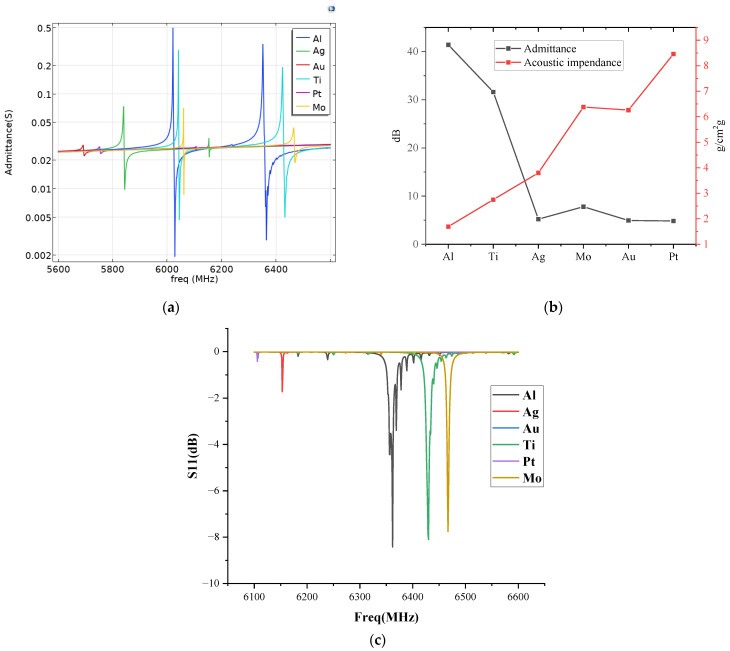
(**a**) Absolute value of the admittance for the electrode materials; (**b**) Relationship between the admittance amplitude and the acoustic impedance of the electrodes; (**c**) S11 reflection spectra of the electrode materials.

**Figure 5 sensors-22-07439-f005:**
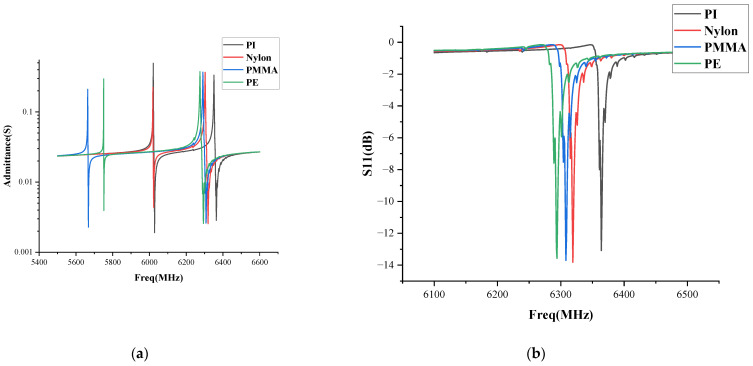
(**a**) Absolute value of the polymer materials’ admittance response; (**b**) S11 reflection spectra of the polymer FBAR.

**Figure 6 sensors-22-07439-f006:**
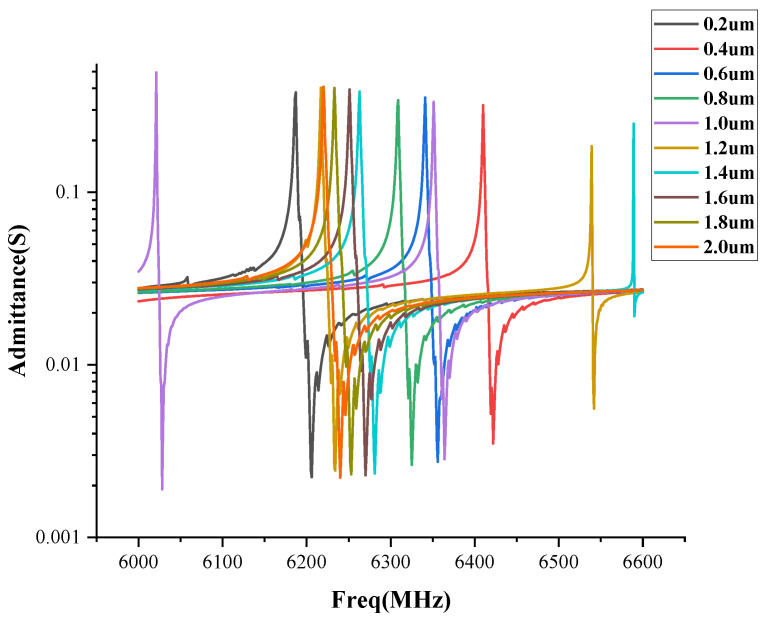
Absolute admittance of the polymer FBAR against the different polyimide thicknesses.

**Figure 7 sensors-22-07439-f007:**
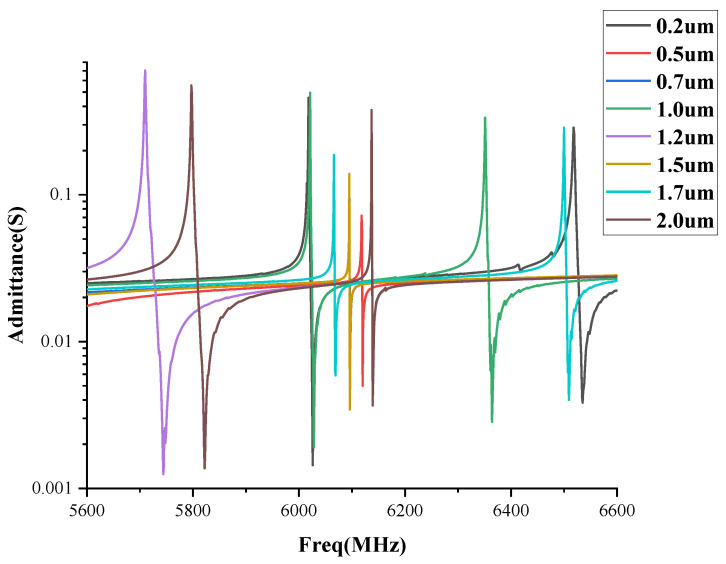
Absolute admittance spectra against the different thicknesses of the Si_3_N_4_ support layer.

**Table 1 sensors-22-07439-t001:** Young’s modulus and Poisson’s ratio of common polymer materials.

Polymers	Young’s Modulus (GPa)	Poisson’s Ratio
PI	3.1	0.34
PA	2.0	0.28
PMMA	3.0	0.40
PE	1.0	0.38

## Data Availability

Not applicable.

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
