# Peer review of "High-Order Harmonic Film Bulk Acoustic Resonator Based on a Polymer Reflector"

_sensors, 2022, doi:10.3390/s22197439_

Round 1

Reviewer 1 Report (New Reviewer)

In this paper, the authors present the numerical results of an FBAR design based on a polymer air cavity. I have a few concerns that should be clarified:

1. The authors state in lines 221 and 222, "These properties remain stable over a wide temperature and frequency range." However, the temperature range used in the study was not presented. And how could temperature affect device response?

2. In section 3.3, the authors discuss the results of the thickness study. It is lacking to present the physical principles that explain the observed effects.

3. It would be important to compare the results obtained with experimental results found in the literature to evaluate their consistency.

Other minor issues are:

4. Was any additional COMSOL module used?

5. It is necessary to increase the resolution and size of the fonts in figures 3, 5, 6, and 7.

6. It is also necessary to provide a review of English, especially in the introduction.

Author Response

Dear reviewer

Kind regards   Mr. Yuxuan Hu

Reviewer 2 Report (New Reviewer)

The paper does not provide information on the size of errors in the obtained results. With what accuracy were the results of the simulation tests obtained? The influence of the system geometry discretization on the final results has not been investigated.

There is also no comparison of the obtained results with the tests of real sets of layers (objects). The obtained results of simulation tests are difficult to compare with the experiments on real objects. Supplementing with such a study would make the work more attractive.

It would be interesting to test for different sizes / shapes of the air cavity.

Author Response

Dear reviewer

Kind regards   Mr. Yuxuan Hu

Reviewer 3 Report (New Reviewer)

There is only theoretical work by making simulations in COMSOL. I suggest to do some measurements on real material samples.

What is the modelling uncertainty?

How to make such samples?

The maximum current and voltage on the sample?

Hoiw is the acoustic efficiency obtained?

Author Response

Dear reviewer

Kind regards   Mr. Yuxuan Hu

Round 2

Reviewer 1 Report (New Reviewer)

All concerns raised during the peer review have been clarified. In my opinion, the manuscript is suitable for publication.

Reviewer 2 Report (New Reviewer)

The authors of the research used CMOSOL simulation software verified by many experts and companies. The obtained results were compared with other literature results to confirm the accuracy of the solutions. In further research, it would be interesting for the authors to take into account the analysis of errors resulting from the selection of discretization steps for the analytical description of the phenomenon under study.

Reviewer 3 Report (New Reviewer)

There is no real sample just theoretical work.

This manuscript is a resubmission of an earlier submission. The following is a list of the peer review reports and author responses from that submission.

Round 1

Reviewer 1 Report

This work, while technically sound, is not really appropriate for a peer reviewed publication.  It is a simulation and exploration of the design space for a particular design, as well as a discussion of the background and results relating to different corners of the design space.  There is no experimental validation, and no new surprising, generalizable, or unexpected theoretical insight into the design and space.  The device is not challenging/impossible to build (which could justify a theoretical exploration, but could also undermine the relevance) - this needs some experimental demonstration to be competitive in the field.

Reviewer 2 Report

Reviewer’s Comments for Author

Manuscript title: High-order harmonic film bulk acoustic resonator based on 2 polymer reflector

Journal: polymers

Present manuscript presents some new interesting information for readers. However, it has some demerits which can be improved. Hence, this work may be accepted after incorporating following corrections-

1.      The novelty of the work should be highlighted to real physics phenomena in both the introduction and abstract. The idea and the results need more depth.

2.      The discussion is sketchy. It is more graphical presentation of various numerical computation, lacks depth and physical contents. The most important results obtained should be clearly highlighted.

3.      Authors should add latest published papers relevant to the present manuscript.

4.      Express the utilities of work done in the manuscript.

5.      Add the following articles to improve the list of references

·         Non-local effect on quality factor of micro-mechanical resonator under the purview of three phase lag thermoelasticity with memory dependent derivative”, DOI: 10.1007/s00339-022-05322-5

·         Analysis of magnetic field effect in micro-beam resonators at distinct boundary conditions”, DOI: 10.1080/17455030.2021.1879407.

6.      Modifications in English language are inevitably required throughout the manuscript. 

7.      Some grammatical and spelling errors degraded the quality of the presentation.

8.      Write the conclusion more precious.

Reviewer 3 Report

This is an interesting paper that presents a design of an FBAR that uses a polymer cavity.  

There are some minor typographical errors that need to be corrected, but then also some further amendments that should be addressed.

 Suggested amendments:

1)     In the introduction, please provide more information on the motivation for a new design before the new design is mentioned, i.e. what problem is the polymer FBAR solving? What does it allow that isn’t already possible? your conclusion mentions “…flexible wearable FBAR sensors.” but this isn’t clear from the introduction.

2)     Your results and discussion section is a simulation and results section there is a separate discussion section. The results introduce the simulation study parameters and is not just results.  I would suggest starting a new section where you say “Simulation analysis is performed….” on page 3 of 12. This section could then introduce the simulation setup, explain what each study will look at and what parameters would be changed. This could then be a new section that includes your results and the setup and could be ‘simulation and results’.  You could consider providing an experimental matrix that shows the fixed and variable parameters for each simulation study.

Other minor corrections/amendments:

3)     Page 3 of 12 - line 90: k2 should be K2.

4)     Page 3 of 12 - line 91: “…blew formula:”   should this be “….below formula:” ?

5)     Page 3 of 12 - line 106: “…large while that of the…”  ‘that of’  can be removed as you say characteristic impedance again.

6)     Page 3 of 12 - line 109: “….support layer to enhances the…” change to “….support layer enhances the…”.  ‘to’ not needed.

7)     Page 4 of 12 -line 122: "pour" should this be "our"?

8)     page 4 of 12 - figure 3: could part b) be put under a) to make the figure larger? also in b) make clear which is which using a label.

9)     Page 5 of 12 - line 143: don’t need the ‘s’ on materials

10)  Page 5 of 12 – section 3.2. Polymer materials: you need to state what the other fixed parameters such as electrode material are for this simulation.

11)  Page 7 of 12 – line 186: says “The 1.5 μm thick polyimide film….” but there is not a 1.5 μm line on the graph? please clarify this.

12)  Page 7 of 12 – line 196-197: “The admittance response amplitude and frequency are higher as the thickness of the Si3N4 layer is between 0.7μm and 1.4μm, …”,  I’m not sure this is obvious when looking at the graph as 1.2 um is higher at lower frequency. Could you annotate the graph and refer to this?

13)  Page 7 of 12 – line 198: “9000m/s”   there should be a space between numbers and units – please check the rest of the manuscript as there are several instances to be corrected.

14)  Page 7 of 12 – line 200: “…and maty induce…”  I am not sure what ‘maty’ means here is this the correct word?

15)  Page 8 of 12 – line 210: “…. the piezoelectric layer to ….” Add a comma after layer “…. the piezoelectric layer, to ….”

16)  Page 8 of 12 – line 214: “polymer FBAR. Resonance.”, extra full stop after FBAR can be removed.

17)  Page 8 of 12 - lines 219-221: polyimide info needs a reference.